# Animal welfare with Chinese characteristics: Chinese poultry producers' perceptions of, and attitudes towards, animal welfare

Qing Yang[1]*, Cathy M. Dwyer[1,2], Belinda Vigors[1], Ruqian Zhao[3], Fritha M. Langford[1,4]

1 Royal (Dick) School of Veterinary Studies, University of Edinburgh, Edinburgh, United Kingdom, 2 Department of Animal Behaviour and Welfare, Scotland's Rural College (SRUC), Edinburgh, United Kingdom, 3 College of Veterinary Science, Nanjing Agricultural University, Nanjing, China, 4 School of Natural and Environmental Science, Newcastle University, Newcastle upon Tyne, United Kingdom

* s1362029@ed.ac.uk

**Data Availability Statement:** This study presents qualitative data from interviews with Chinese poultry producers within the authors' professional network. All participants have given written consent

## Abstract

China's poultry industry faces challenges in adopting and sustaining cage-free systems for poultry production. Effective interventions are crucial to support producers transitioning from cages to alternative systems or maintaining cage-free systems to improve animal welfare. However, little is known about how Chinese poultry producers perceive animal welfare in relation to cage-free systems and the importance of animal welfare in poultry production. Through a qualitative interview study with 30 Chinese farm owners, managers and senior managers from large-scale egg and broiler farms using cages and non-cage systems (collectively referred to as "producers"), this paper explores Chinese poultry producers' attitudes and perceptions regarding animal welfare and welfare in different poultry housing systems. Template analysis was used to analyse the data from semi-structured interviews, which generated themes related to the participants' awareness and understanding of the concept of animal welfare, the factors that impacted their choices of different housing systems, and the perceived priorities in poultry production. The responses revealed that the participating producers had a strong awareness and knowledge of animal welfare. However, the participants' understanding of the term is heterogeneous: generally, egg producers emphasised natural behaviours, whereas broiler producers prioritised health and productivity. Nevertheless, profitability, leadership, and organisational policies primarily influenced housing system choices rather than animal welfare values. Economic motives drove egg producers towards cage-free systems, prompted by consumers' and companies' demand for cage-free eggs committed to transitioning away from cages by 2025. In conclusion, tailored interventions for different poultry sectors within China are necessary. While animal welfare values matter, economic incentives seem more promising for steering the shift towards and maintaining cage-free poultry production.

to use direct quotes from their responses in research papers and QY's PhD thesis. However, publishing the interview transcripts, as the PLOS ONE research data policy suggests, is not feasible. Despite anonymising all transcripts, the specific information discussed in the interviews—such as production systems, flock size, and indigenous chicken breeds—poses a risk of identifying individual participating producers. Additionally, publishing anonymised transcripts would violate the obtained written consent from participants, thus non-compliance with the UK General Data Protection Regulation (UK GDPR) and the Data Protection Act 2018. Due to these restrictions, the authors will only provide the interview guidelines as supplemental material to this paper. A summary of all utilised quotes supporting the themes in Chinese will be available upon request from the Human Ethical Review Committee via email at herc. vets@ed.ac.uk.

**Funding:** Dr C.M.D. and the Royal (Dick) School of Veterinary Studies at University of Edinburgh received funding for this project through Open Philanthropy (https://www.openphilanthropy.org/). There is no grant number for this award. The funder had no role in the study design, data collection and analysis, publication decision, and manuscript preparation.

**Competing interests:** The authors have declared that no competing interests exist.

# 1. Introduction

Over the past four decades, the poultry industry in the People's Republic of China (hereafter referred to as China) has undergone substantial intensification and industrialisation. This transformation has shifted from traditional backyard farming to larger, more intensive production methods to enhance capacity and efficiency [1, 2]. Conventional or battery cages have become a hallmark of this industry [3], as nearly 90% of laying hens are estimated to be raised in these cages [4], and there is a growing trend of using similar systems for broilers [5, 6]. China's poultry sector is marked by its intensive nature, with the chicken industry alone producing 29.7 million tonnes of eggs and 15.4 million tonnes of meat in 2021, propelling China to become the world's leading poultry producer [3, 5, 7]. This trajectory of intensification and industrialisation will persist in meeting the escalating demand for eggs and poultry meat from the country's vast population [3, 8].

Contrastingly, the European Union (EU) has witnessed a shift away from conventional cages to alternative housing systems for poultry driven by scientific insights and public concerns for animal welfare [9, 10]. Broilers in the EU are commonly reared on littered floors, with cages being uncommon [11, 12]. While all poultry production systems possess distinct advantages and disadvantages [5, 13], well-managed cage-free systems hold greater potential to ensure poultry welfare [5, 14–16]. Cages fail to provide the space and environmental complexity necessary for chickens to express highly motivated behaviors like dustbathing, perching, and foraging [5, 14–17] which are integral to welfare [15, 16]. Consequently, EU legislation prohibits the use of conventional cages and exclusively permits enriched cages and cage-free systems (e.g., single-tier, multi-tier and free-range systems) in the egg industry [18]. In addition, the European Commission is proposing a legislative phase-out of all cage systems, including enriched cages, for diverse farm animal species, including laying hens and broilers [19]. Dozens of global manufacturers, hotels, supermarket chains, and food service companies have committed to sourcing cage-free eggs throughout their international operations, initially focusing on European and North American markets [20]. In this context, scientific insights, legislation, and market dynamics in the EU are synergising to steer the transition toward cage-free housing systems for farm animals [21].

In recent years, while cages have been the dominant production system, the adoption of cage-free systems in the egg industry has begun to grow in China. This transition is driven by consumer preferences and businesses' growing demand for cage-free eggs [5, 22, 23]. Local chicken breeds have historically been raised in small-scale free-range systems [24], and some Chinese consumers display a willingness to pay for free-range eggs and meats [25, 26]. However, there is limited information regarding consumer egg purchases from other cage-free systems [23]. Beyond consumers, over 70 international food businesses, retailers, and hotel chains have pledged to source exclusively cage-free eggs in the Chinese market [20]. Yet, the scarcity of cage-free egg supply poses challenges for companies to meet their cage-free commitments [27]. Moreover, observational evidence shows that the broiler sector has witnessed a shift from floor-based systems to cages [28]. Therefore, initiatives to expedite and sustain cage-free systems in the poultry industry are imperative for the welfare of billions of chickens in China [7].

Producers wield direct influence on animal welfare by determining the adoption and execution of animal welfare initiatives [29–31]. Research on European producers demonstrates that their inclination to choose higher welfare housing systems and engage in welfare schemes correlates with their personal values and attitudes toward animal welfare [32–35]. Those defining animal welfare in terms of biological functioning and productivity generally favour conventional production systems [34–36] and farmers who prioritise the importance of natural

behaviours for animals choose production systems that allow behavioural expression [32–35]. In addition, Western producers' decisions regarding production systems may be influenced by diverse perspectives on ethical animal treatment. For instance, some free-range farmers advocate for free-range systems, believing they enhance animal welfare [37], while others argue that cages are preferable for laying hens, citing improved health benefits [38]. Moreover, factors such as business benefits associated with animal welfare [39], operational priorities and challenges [30, 33, 35, 36, 40] and varying degrees of significance placed on animal welfare in production, influenced by intrinsic and extrinsic factors [35, 41], also play pivotal roles in shaping producers' choices of housing systems. Nevertheless, producers' decisions regarding housing systems can deviate from their perceptions and attitudes because their choices are also linked to country-specific dynamics (e.g. culture and religion) [30, 39], sector-specific considerations (e.g. different species and their needs) [42, 43], and farm-specific contexts (e.g. leadership and organisational culture) [44–46]. Therefore, understanding personal and contextual factors is a crucial step in improving animal welfare [47] within Europe and across Asian nations [30, 39].

However, previous research predominantly stems from Western contexts, leaving a knowledge gap regarding Chinese producers' perceptions and valuations of animal welfare [44]. There is limited insight into producers' perspectives on the term "animal welfare" in large animal operations, and definitions are often derived from Western benchmarks [44, 48–50]. A limited number of quantitative studies have surveyed diverse Asian animal agriculture stakeholders. Participants exhibited varying degrees of intention to enhance animal welfare [49, 51], prioritised distinct animal welfare issues [30, 43], encountered varied challenges [49], and proposed context-specific solutions for animal welfare improvement [30, 52]. In addition, qualitative investigations have recently delved into farm workers' understanding and attitudes towards animal welfare in the Chinese dairy [46] and aquaculture sectors [42]. These studies revealed that participants in different job roles had divergent levels of animal welfare awareness, accompanied by varying degrees of significance placed on animal welfare in practice, and leadership support played a pivotal role in enhancing animal welfare [45, 53]. Therefore, it is important to explore business leaders' understanding of animal welfare to ensure effective strategy formulation [53].

This paper explores Chinese poultry business leaders' knowledge and definitions of "animal welfare", their choice of housing systems concerning animal welfare, and their perceived importance of animal welfare in poultry production. Understanding farming business leaders' perspectives and priorities concerning poultry welfare can contribute to designing strategies that address the stalling progress of cage-free systems in egg production and growing concerns about cage utilisation in the broiler sector in China.

## 2. Materials and method

### 2.1 Sampling

This study chose a qualitative approach, applying semi-structured interviews, to explore producers' complex perceptions and attitudes toward animal welfare. Purposive and snowball sampling techniques were used to enable the selection of a population of interest. The population targeted for this study were business leaders from medium- to large-scale commercial laying hen and broiler farms in mainland China, as they were in the decision-making positions for selecting housing systems. Most participants were located in China's top poultry production regions: Shandong, Hebei, Henan, Liaoning, Jiangsu, Sichuan, Anhui, Hubei, Heilongjiang, and Jilin. Based on the literature [4, 8] and the lead author's industry-based knowledge, the scales of targeted farms were defined as a minimum of 50,000 birds and 5,000 birds in annual stock for cage and cage-free egg farms respectively, and 2000 birds

produced annually for cage and cage-free broiler farms. The sample consisted of producers using cages (conventional battery cages) and cage-free systems, including single-level systems (for laying hens), multi-tier systems (for laying hens), floor-based systems (for broilers), and free-range systems. Conventional laying cages typically comprise small enclosures with sloping floors made of welded wire mesh [54]. In single-level systems for laying hens and floor-based systems for broilers, birds are housed on the floor, often covered with litter. Multi-tier systems are similar to single-level setups but incorporate multiple tiers or platforms, enabling hens to utilise the vertical space within the building [55]. Free-range systems involve housing birds in either single or multi-tier setups while providing access to outdoor areas [55]. Considering some poultry farms were transitioning between cages and cage-free systems, producers using cages and cage-free systems concurrently were also included in the sample.

The study involved 30 participants with 15 egg producers and 15 broiler producers. We recruited participants through diverse methods, including WeChat advertisements (a popular Chinese social platform, equivalent to WhatsApp) and referrals from existing participants and industry associates. Recruitment stopped once no new information emerged [56]. The participants comprised professionals in different roles within farming companies, including farm owners, managers, and senior executives who oversee one or multiple farms in large integrated animal production firms. For brevity and simplicity in the following sections, these individuals are collectively referred to as producers. Information regarding farms' locations, scales and housing systems can be found in Tables 1 and 2.

**Table 1. Egg farms' locations, scales and housing systems.**

| Participant | Locations | Farm scales [a] | Production systems |
|:---:|:---:|:---|:---|
| 1 | Sichuan | 1 million | Battery cage |
| 2 | Nationwide | 13 million | Battery cage |
| 3 | Hebei | 12,000 | Free-range |
| 4 | Shanxi | Cage: 350,000 | Battery cages |
| | | Multi-tier:40,800 | Multi-tier |
| 5 | Sichuan | 20,000 | Free-range |
| 6 | Nationwide | Cage: 110,000 | Battery cages |
| | | Multi-tier: 25,000 | Multi-tier |
| 7 | Nationwide | Cage: 300,000 | Battery cages |
| | | Single-tier: 10,000 | Single-tier |
| 8 | Henan | 500,000 | Battery cage |
| 9 | Guangdong | 1.2 million | Battery cage |
| 10 | Nationwide | 300,000 | Free-range |
| 11 | Henan | 40,000 | Free-range |
| 12 | Beijing | 100,000 | Free-range |
| 13 | Nationwide | Cage:30 million | Battery cages |
| | | Multi-tier:200,000 | Multi-tier |
| 14 | Fujian | Cage: 1 million | Battery cages |
| | | Single-tier: 100,000 | Single-tier |
| 15 | Nationwide | 100,000 | Single-tier |

[a] Farm scales = approximate number of birds kept annually per farm.

**Table 2. Broiler farms' locations, scales and housing systems.**

| Participant | Locations | Farm Scales [b] | Production systems |
|---|---|---|---|
| 16 | Nationwide | 60 million | Cages |
| 17 | Shandong | 40–50 million | Cages |
| 18 | Shangdong | 3.5 million | Cages |
| 19 | Shandong | 4.6 million | Cages |
| 20 | Shandong | 12–14 million | Cages |
| 21 | Shandong | 2 million | Cages |
| 22 | Shandong | 150 million | Cages |
| 23 | Shandong | Cages: 50 million | Cages |
|  |  | Floor-based:70–80 million | Floor-based |
| 24 | Nationwide | 323 million | Floor-based |
| 25 | Nationwide | 520 million | Cages |
| 26 | Guangdong | Cages: 1.88 million | Cages |
|  |  | Floor-based: 3 million | Floor-based |
| 27 | Guangdong | 100–120 million | Cages |
|  |  | Floor-based: unclear | Floor-based |
| 28 | Hebei | 30 million | Cages |
|  |  | Floor-based: 20 million | Floor-based |
| 29 | Guangdong and Guangxi | 80 million | Free-range |
| 30 | Shandong | 46 million | Floor-based |

[b] Farm scales = approximate number of birds produced annually per farm.

## 2.2 Interviews

Between the 6th of April and the 10th of July 2021, the lead researcher (QY) conducted semi-structured interviews over the phone. Two interview guides were developed for interviewing egg producers (S1 Appendix) and broiler producers (S2 Appendix). The interview questions sought to explore the participants' understanding of animal welfare, perceptions of cage-free systems, the importance of animal welfare in poultry farming, and their impressions of laying hens and broilers. Additional probing questions were asked to understand participants' experiences of using both cages and cage-free systems. The interview guidelines were developed in English to facilitate all authors' contributions and translated into Chinese for use in interviews. Conversations were mainly driven by the participants' responses to facilitate open discussion. All interviews were audio recorded (Mi 10, Xiaomi Communication and Technology Ltd., China) and transcribed into written Chinese by QY. Transcripts were uploaded and analysed in Nvivo (QSR International Pty Ltd. Version 12). QY translated the chosen quotes into English to facilitate the presentation of the results in this paper because one translator working with the data can maximise the reliability of translation and interpretation [57]. A bilingual researcher with expertise in a similar research discipline was provided with selected quotes, specific codes, and relevant text fragments to ensure an understanding of the interview context. Subsequently, the researcher reviewed the translation of the quotes, codes, and themes to confirm that the intended meaning and contextual nuances in Chinese were accurately preserved in the translations. In the presentation of quotations, some contextual information was added in square brackets to ensure clarity.

## 2.3 Data analysis

Template analysis was used to analyse all data. Template analysis is a form of thematic analysis which enables a structured approach to data coding with the flexibility of adapting it to the needs of a study [58], which helps understand a relatively under-explored topic [59]. It has been used to analyse qualitative data collected from focus group discussions and semi-structured interviews with farm animal producers from different countries and cultures, including China [45, 46, 59].

QY followed the typical steps of applying template analysis described in King and Brooks [60]. QY re-read the interview transcripts and examined their accuracy against the recordings to become familiarised with the data. Several themes corresponding to the study questions were defined in advance (e.g. "definitions of animal welfare" and "perceptions of chickens"). Before preliminary coding, QY selected ten interviews containing rich data relevant to the research topics and coded a text segment with a phrase under a theme. For instance, the codes "high egg production" and "healthy birds" are put under the parent code of "biological functioning" under the theme of "definition of animal welfare". QY used the list of codes and themes drawn from the ten interviews as a template and applied it to the rest of the data sets. Codes and themes were refined and divided, and new codes were added during the process.

Since some participants accounts of experience tended to be fragmented in the interviews, QY kept summaries for each interview, to capture the participants' holistic experiences and perspectives [60]. QY created an audit trail during data analysis which kept dated versions of the template, including reflective comments on coding choices and why particular modifications were made.

## 2.4 Ethics

Ethical approval for the interviews was sought and granted by the Human Ethical Review Committee of the Royal (Dick) School of Veterinary Studies at the University of Edinburgh (code HERC_655_21). Before each interview, participants reviewed a participant consent form and introductory information to the research project, which specified that participants' identities would be anonymised and that they could withdraw from the study discretionarily. Written consent was obtained from the participants before the interviews.

## 3. Results

This section describes Chinese poultry producers' perceptions of, and attitudes towards, animal welfare and their evaluations of, and decisions on, different housing systems concerning animal welfare. The first two sub-sections present Chinese poultry producers' knowledge and definitions of animal welfare. The subsequent sub-sections detail producers' choice of housing systems in relation to animal welfare values, and the last sub-section describes the perceived importance of animal welfare in Chinese poultry production.

### 3.1 Participants' awareness of the term animal welfare

**3.1.1 Familiarity with the term.** To investigate Chinese producers' perceptions of, and attitudes towards, animal welfare, confirming that the participants know the term "animal welfare" is essential. The findings show that all the participants have heard of the phrase "dongwu-fuli" (动物福利) (i.e. the Chinese translation for the term animal welfare). In some egg producers' narratives, this term arose naturally in the discussion of cage-free housing systems and cage-free egg production. More specifically, the word "fuli" (福利) (welfare) was often used interchangeably with "cage-free" in egg producers' statements. For example, "welfare

eggs", "welfare shed", and "welfare farming" were equivalent phrases to "cage-free eggs", "the shed using cage-free systems" and "cage-free production". Therefore, it is evident that "welfare" is not only a familiar term but is also commonly used in egg production. In contrast, the broiler producers did not use the term "animal welfare" or "welfare" to describe their practices or experiences in broiler farming:

"We hope to develop the market [for cage-free eggs]. . .and aim to produce affordable welfare eggs."

(P15 cage-free egg producer)

"We have several sheds on the farm, and one [of them] is a welfare shed."

(P7 cage and cage-free egg producer)

"Cage-free production means that the chickens can live freely and move freely. They are treated with higher welfare and can enjoy the freedom of expressing their nature; that is what we call welfare farming."

(P11 free-range producer)

Two producers acknowledged that they could not articulate what animal welfare meant but were aware of some higher welfare husbandry practices:

". . .Animal welfare. . .I heard them talking about it, but I am still unclear what it is. . .our chickens' beaks are trimmed, but welfare doesn't do beak-trimming. I heard them mention some practices [with higher welfare] but did not get the definition."

(p4 cage and cage-free egg producer).

"Animal welfare, I don't know how to explain it. There are animal-based indicators. . .in general, they have a better housing environment with more space and some choices. . .I don't know how to define. . .and explain the concept in detail."

(P27 cage and cage-free broiler producer)

Therefore, it was evident that all the producers had heard of the term animal welfare and had some knowledge of this concept. Amongst participants, egg producers more commonly used animal welfare than boiler producers.

**3.1.2 Western influences.** Several producers followed Western practices when discussing how the farms started cage-free poultry production. Two producers explained how they began their cage-free operations after visiting poultry farms in Europe:

"30 years ago, we learnt that . . .Europe started to use housing systems with higher welfare. . . I visited Europe several times to see the operations. In 2010, we introduced their equipment into China and set up an automated [cage-free] egg farm."

(p14 cage and cage-free egg producer)

"We followed the European example to design and build our floor-based broiler farms because Europe stresses the importance of animal welfare, they don't want to use cages. . . a big group of us went abroad to learn [about the system], and the European practice influenced us."

(P16 cage broiler producer)

Although some cage-free producers began their cage-free poultry production based on learning from their Western counterparts, this does not mean the Chinese producers did not know what good cage-free practice entailed before they came across the term. This is reflected in one free-range producer's narrative when she recalled her experience of interpreting the word:

"When I first learned the term animal welfare, it sounded so impractical. . .because we often talk about human welfare. . .and when human welfare issues haven't been solved, we are solving animal welfare issues?. . .it sounds so westernised. . .but later. . .I realised it re-phrases our free-range production standards . . .animal welfare farming [practices] and the Five Freedoms. . . are the basic living needs."

(P5 free-range producer)

Therefore, while Western practices influenced some participants' learning of modern cage-free poultry production, other cage-free producers could relate their farming experiences to animal welfare and incorporate their understanding of animal welfare into their daily practices.

## 3.2 Conceptualisations of animal welfare

Participants generally approached the concept of animal welfare from a positive perspective, that is, what constitutes good animal welfare, rather than focusing on indicators of poor welfare. In particular, participants' understanding of animal welfare can be broadly grouped into three categories: good health and productivity, positive affective state and providing opportunities for behavioural expression.

**3.2.1 Good health and productivity.** When defining animal welfare, good health and productivity emerged as the most important indicators for the majority of broiler producers and some egg producers. As one broiler producer described:

"Animal welfare means eating, drinking, and providing a good living environment. If animal welfare is poor, they won't grow well, their physical health declines, production efficiency reduces, the birds are more likely to get sick, and their growth rate slows. [animal welfare] is not limited to the freedom of movement. . .ultimately the excellent production performance, or efficiency, is the most convincing [indicator of good animal welfare]."

(p 22 cage broiler producer).

Similarly, a senior manager of a large integrated egg production group stated:

"Animal welfare essentially is to maximise chickens' production performance and provide an appropriate living environment, so that they can provide products with maximal efficiency."

(p13 cage and cage-free egg producer)

**3.2.2 Positive affective state.** Several producers emphasised the importance of ensuring a neutral or positive affective state to enable good welfare. For them, a neutral state, such as "no stress", as well as a positive affective state, such as "happy" and "comfortable", were all used as indicators of good welfare:

"...We are willing to provide more animal welfare to the broilers. In production, animal welfare mostly refers to reducing stress. No stress makes chickens more comfortable."

(p24 cage-free broiler producer)

"Animal welfare is to let the chickens live a happy life."

(P7 cage and cage-free egg producer)

"[Good animal welfare] should make the chickens feel comfortable."

(P23 cage and cage-free broiler producer)

**3.2.3 Providing opportunities for behavioural expression.** Nearly all the egg producers and a few cage-free broiler producers believed free movement is crucial for good animal welfare. Some of them defined animal welfare as "freedom", "not restricted", "return to natural living", and the ability to move freely:

"Animal welfare is to give [chickens] the rights of freedom."

(P5 free-range producer)

"We typically understand animal welfare as chickens are not restricted. .. .. .and she can restore or live a natural life as much as possible."

(p10 free-range producer)

"Animal welfare means chickens can return to natural living."

(P11 free range producer)

"The ability to move freely is primary for the animal's welfare."

(P6 cage and cage-free egg producer)

In contrast, some producers did not see the birds' abilities to express natural behaviours as necessary or essential due to the genetic and dispositional traits of white-feathered broilers and laying hens. They argued that the modern breeds of chickens had been genetically selected for less movement and inhibited behavioural expression, and hence, they did not need to express specific behaviours:

"The caged layers have been genetically bred for many years, and they are not suitable [for cage-free systems]. Not all of the chickens. . .like to fly around."

(p13 cage and cage-free egg producer)

"White broiler is bred for fast growth and high yield of meat. This is the ultimate purpose, and they are genetically bred to stay still, grow faster and eat more. . . the birds are not bred for running faster or preferring movement."

(p25 cage broilers)

**3.2.4 Animal welfare as a holistic concept.** Several producers stressed that animal welfare should be a holistic concept, and good animal welfare must meet animals' different needs and go beyond the farming stage. Particularly, animal welfare needs to be considered from the

animals' perspectives not only in production but also in genetic breeding, depopulation, transport and slaughter:

> "Even after the flock is depopulated, the process of catching and slaughtering also involves animal welfare. . .so it's a broad topic . . . [animal welfare] is systematic, it does not only relate to the farming stage but also include transport, bird's placement. . .it should be holistic."

> (p22 cage broiler producer)

> "Genetic breeding is not [good for] animal welfare. Why? The heavier the broilers get, the more pressure their leg joints are under. Why are there so many leg problems in fast-growing breeds? It is because of being overweight."

> (p16 cage broiler producer)

Overall, participants could explain their understanding of animal welfare at the farming stage as well as in breeding, transport and slaughter practices. These findings suggest that participants had strong animal welfare knowledge.

## 3.3 Interconnections between animal welfare values and housing systems

It was found that the producers' values and choices of housing systems were interconnected. Most producers' values appeared to underpin their choice of housing systems. However, some producers' values are not necessarily consistent with the choice of housing systems.

**3.3.1 Consistency between values and choices.** Most producers who valued good productivity, health, and low stress in a highly controlled environment considered cages superior to a cage-free or outdoor setting. For them, the birds' abilities to express appropriate behaviour are either unnecessary or less important. As such, they strongly supported cage systems, which was consistent with their values:

> "Cages are highly cost-effective. They provide better animal welfare, [because birds] don't get sick, use fewer drugs, have higher survival rates; the environment is highly controllable, and production efficiency is higher."

> (p25 cage broiler producer)

> "The [caged] shed is very clean, and you don't need to disinfect it much. You control the environment well and don't disturb the chickens, providing them an environment where they can grow comfortably and freely, without stress."

> (p17 cage broiler producer)

Similarly, reflecting on the connection between values and housing systems, nearly all egg producers and a few cage-free broiler producers believed that meeting animals' behavioural needs is essential for good animal welfare. Hence, they chose cage-free systems:

> "Chickens (should) live in a good and comfortable environment. . . Living in the [caged] environment, they are not likely to be mentally healthy, cage-free systems are much better."

> (p10 cage-free egg producer)

"Cages affect animals' nature, in five freedoms of animal welfare, the primary freedom is movement, [otherwise] it's against animals' nature."

(P29 cage-free broiler producer)

Therefore, most producers believed their chosen housing systems benefited animal welfare, and their choices appeared to reflect their animal welfare values.

**3.3.2 Inconsistency between values and choices.** Inconsistency between participants' values and choices of housing systems was also seen in the participants. The inconsistency is mainly caused by external factors, including profitability and the company's leadership decisions. For instance, although some cage producers agreed that behavioural expression was important and cage-free systems provided better welfare, making a profit and surviving market competition were the key factors preventing them from using cage-free systems:

"From the animal welfare perspective, cage-free systems are certainly better than cages. . .but the key is, from the economic profit perspective, when there are no [legal] requirements, and you improve animal welfare by investing more, it is not cost-efficient. We want to choose a cheaper housing system because we are a rational company."

(P1 cage egg producer)

"From an animal welfare perspective, cage-free systems are preferred, [because] birds are given more space and freedom. . . For large-scale egg producers, profit is higher in cage systems than in cage-free systems. However, we can't even maintain a high profit from cage systems. . .so we will not consider cage-free systems. . .which bring more uncertainties and higher deficit. . .."

(P9 cage egg producer)

The leadership's decisions in companies also played an essential role in adopting cage-free systems, especially for the farms established by owners and companies. Although several cage-free farm managers personally did not equate good animal welfare with a cage-free or "natural" environment, the choice of production systems was determined by the farm owners and companies' leadership, hence irrelevant to the participants' values:

"The farm owner wanted to produce some more natural and high-end agricultural products. . .[because] he believed there was a market for products of better safety and quality."

(p12 free-range egg producer)

Another cage-free egg producer followed his senior management's decision to set up a cage-free egg farm to explore the feasibility of cage-free egg production in China:

"After the EU directive took effect in 2012, our [European] headquarter office wanted to see if there is similar demand in China [for cage-free eggs] . . . so we want to plan early and explore [the market] . . . we followed the EU standards and built a shed to study its costs and viability [of producing cage-free eggs]."

(P7 cage and cage-free egg producer)

Therefore, while some producers' values were consistent with their housing systems, others did not align with the choices. Instead, profitability and leadership decisions led to the selection of housing systems.

### 3.4 Attitudes towards animal welfare

**3.4.1 Animal welfare as a non-priority.** In general, consideration of animal welfare was not perceived as necessary in poultry production. Exceptions were a couple of cage-free producers working for leading agriculture groups. They related animal welfare to the companies' missions and social responsibilities and believed good animal welfare was an indicator of a good company and food safety:

> "Our company's vision is to contribute to food safety in China. . .if a company promises to produce products with higher welfare. . .the company can't be bad. . .only a good company can promise to adopt [higher animal welfare] practice."

> (p15 cage-free egg producer)

> "I think the motivation and social responsibility of running an agricultural business should be providing better food choices for consumers. . .animal welfare is human welfare. . .producers should take on the social responsibilities."

> (p11 free range producer)

Nevertheless, regardless of their views on animal welfare, most cage-free producers did not choose cage-free systems for animal welfare reasons. Instead, the choice rationale came from meeting the perceived consumer demand and profitability. One cage-free egg producer clearly stated:

> "The cage-free eggs might have a better prospect, [because] raising the conventional chickens probably can't make much money and cage-free eggs might be sold at a better price and [producers] can make some profit. . .I will try other cage-free systems if they are cost-effective. . .whichever is profitable, I will use it."

> (p4 cage and cage-free egg producer)

In the white-feathered broiler sector, animal welfare was generally considered unimportant. Nearly all the participants had started or completed transitioning from floor-based systems to cages. The transition was driven by profitability and market demand for affordable meat, even though the economic investment is much higher in cages than in floor-based systems. One cage broiler producer observed how animal welfare was less and less emphasised since the transition began:

> "So far animal welfare is less and less considered. . .we increasingly realise that cage systems make more money indeed. . .from the perspective of profit, [the producers] will not consider animal welfare. . .Since 2013, animal welfare in white-feathered broiler production has been paid less attention."

> (p16 cage broiler producer)

Other broiler producers explicitly stated their indifference to animal welfare:

"At present, almost only the international companies are still using floor-based systems to raise broilers. . .because of the policies in those companies. . .Europe and America have higher welfare requirements, we Chinese have no faith [in animal welfare], don't we?. . . Animal welfare is not important to us. It is like growing wheat and peanuts. Just grow them. You don't complain about the planting density of wheat, do you? crops are crops."

(p20 cage broiler producer)

"As a business, [our goal] is to turn them into protein as fast as possible. We don't consider things like [animal welfare] too much".

(p26 cage and cage-free producer)

In addition, animal welfare was regarded as conflicting with food security and human welfare. Producers widely agreed that animal welfare should not be considered when securing food supply for mass consumption was a priority:

"For high-end [products], animal welfare is important. For the low-end products for mass consumption, animal welfare is not considered. After all, it is based on cost-benefit analysis."

(p26 cage and cage-free broiler producer)

"It is hard to define whether or not animal welfare is important. . .for companies like us, we think it is important, for other companies. . .they need to secure the food supply, then [animal welfare] is not important."

(p10 free-range producer)

Therefore, producers perceived caring about animal welfare as incompatible with profitability, food security and human welfare. As businesses, consideration of animal welfare was not considered necessary in poultry production, especially broiler production, unless the company's business orientation targeted the market of higher welfare products.

**3.4.2 Profitability as a priority.** The emphasis on profitability was ubiquitous in the participants' narratives. A few producers believed that profitability was an inherent feature in farm animal production and that improving animal welfare offsets the pursuit of efficiency and profitability. Thus, it is undesirable for Chinese farming businesses to use cage-free systems. As such, the "pressure" of gaining profits was often highlighted as a significant reason for not choosing cage-free systems or shifting from floor-based systems to cages in the broiler sector:

"We are doing commercial animal production, not raising pets or exhibiting animals, so [the farming practice] needs to consider the nature of animal farming. . . [using cage-free systems] brings a massive increase in costs. In the short term, I don't think it is realistic in China."

(p23 cage and cage-free broiler producer)

"For the industry of layers. . .there are almost no profits. . .so we always use cage systems with high stocking density, because only this system can produce the most eggs with the least investments, and we can barely survive [in the market] . . . most [producers] do not like animal welfare."

(p13 cage and cage-free producer)

"Profits are the major consideration when we transition to cages. This is the biggest factor. . .because the investment in converting to cages is large. . .if we can't recover the costs soon, we wouldn't have changed. It is all about profitability."

(P21 cage broiler producer)

In addition, pressure was imposed on the business leaders, as they were responsible for economic profits for the investors, and broiler production is a profit-oriented business:

"It was tiring. . .not everyone can withstand that level of stress. . .if the production performance was poor, the investors thumped on the table at me. . .I was so embarrassed. . .they invested billions in the business. I will be in trouble if the economic returns are not good."

(p16 cage broiler producer)

"A broiler farm requires . . .nearly one hundred million of investment. What is the purpose of the investment? It can't be losing money, can it? The purpose is to make a profit, and nobody will do it if there is no profit."

(p8 cage broiler producer)

Similarly, profitability drove the yellow broiler sector from free-range systems to cages, and the transition does not consider animal welfare. Yellow-feathered broilers are often raised in free-range systems for better meat taste which consumers prefer over white-feathered chickens [61, 62]. These broilers are commonly sold in the live poultry market for family consumption. However, the live poultry trade in wet markets has been banned in some regions due to the impact of COVID-19 pandemic and Avian Influenza, thus processed chicken meat sales have been growing [62]. Cage-raised broilers have a better carcass appearance and thus are favoured by the producers. As one participant stated:

"In recent years. . .we tend to use cages. . .live poultry trade is increasingly hard in big cities. . .we will sell the frozen chicken meat to consumers, just like the while-feathered chicken meat production. . .in the Chinese market, animal welfare is not stressed."

(p27 cage and cage-free broiler producer)

**3.4.3 Animal welfare in a Chinese context.** Several producers emphasised that improving animal welfare needs to consider a Chinese context. They further explained that the Chinese context differs from the West: producers are under more pressure to control animal diseases and provide sufficient animal protein for a large population with limited resources. Hence, cages, which economise resources the most, fit in China's national context and are a way of improving animal welfare with Chinese characteristics:

"A Chinese context is that there are many producers, and their farming conditions differ vastly. The disease control is not as good as it is in Western industrialised, intensified, and standardised farms . . .what works well abroad does not necessarily work in China. . .if we adopt the foreign model of floor-based systems, we will need much land. . .the scale of production in floor-based or free-range systems is not enough in China. . .the number of farmed animals and a large population in China determine that a suitable housing system in China should provide chickens with welfare, ensures lower production costs and [good]

biosecurity. Hence, we adopt this multi-level welfare farming system, [i.e., the cages] . . .it is called a [higher] welfare farming model with Chinese characteristics."

(p17 cage broiler producer)

"China is different from the Western countries: the population is larger, and the land is scarcer. In this case, if [we] want more output per unit area, we must adopt cages. . .the essential difference between China and overseas is that they have almost finished their industrialisation process. . .and can meet their demand for animal protein. Still, the Chinese per capita consumption of animal protein is far from sufficient. . .and at this historical stage, we also have the challenges of land resources. . .and raw materials of crops, so we have to pursue higher efficiency. . . we are facing different developmental challenges."

(p25 cage broiler producer).

Apart from supporting cage systems, several producers from large poultry operations advocated large-scale intensive poultry production as the best poultry farming practice that suits a Chinese context:

"Indsutralisation and intensification of the layer industry is a good outcome; after all, China is a developing country, and we just solved the problem of food shortage. . .if we convert to the farming systems with higher welfare. . .at least the [egg] price will double, and it will affect many people's food supply"

(p13 cage and cage-free egg producer)

"Industralisation and intensification is good for broiler production, just like factory farming, because the production cost is low."

(p20 cage broiler producer)

In summary, to some producers, improving animal welfare must consider the Chinese context, in which conventional cages and intensive farming practices are favourable because they can secure food supply and are cost-efficient.

**3.4.4 Opposition against Western welfare standards.** Compared to egg producers, broiler producers demonstrated stronger opposition against the Western animal welfare standards because they promote floor-based systems to improve animal welfare. They believed floor-based systems are old-fashioned and should be abandoned:

". . .Some experts believed in a traditional view of welfare: one must use cage-free systems, animals must have space to express their nature, and the environment must be enriched. . .at present, based on the Chinese context and farming conditions, I believed [cages are] the higher welfare housing systems that suit China. So, we don't follow the Western requirements, such as providing space of movement. . . they don't necessarily work in China."

(P17 cage broiler producer)

"I have been back from the United States for seven years [from my study], and my biggest take is that the foreigners' practices [of using floor-based systems] are obsolete. . . . . .and they don't want to. . . . . .embrace new things. . .we will never go back [to floor-based systems]."

(P25 cage broiler producer)

Facing the promotion of cage-free housing systems, these producers saw Western animal welfare standards as an imposed Western ideology and a political tool to restrict the development of the Chinese poultry industry. Therefore, they aspired to change the "rules of the game" and start a "revolution". As two cage broiler producers elaborated:

"Animal welfare is the largest hypocrisy in farming practices from the West. For example, dairy welfare is not stressed [in the West] because they have a much greater demand for dairy products than China. . .so they don't talk about dairy welfare much or do very little about it. Why do they stress poultry welfare? Because their poultry production can meet their own needs, now they want to restrict the development of fast-growing countries like China because China's poultry production efficiency has exceeded the West. . .in the future, they possibly use the established rules to restrict us. If they don't restrict us, in the future, we might restrict them. . .as late mover companies, we are facing [the challenge of] breaking through the Western logic or the industry standards set under the Western discourse power. . . We are growing stronger and stronger, and we challenge the West's logic and discourse power as we have our own developmental needs. The theory [of animal welfare] has no right or wrong, but the country that seizes the discourse power will develop better and win the competition."

(p25 cage broiler producer)

"[Cages are] a welfare farming model with Chinese characteristics. It is like developing socialism with Chinese characteristics. . . cages are suitable for the future of Chinese agricultural development. Chinese agriculture [industry] is big but not strong now, so we need to grow the industry 'big and strong'. . .this is a revolution. . .we are leading a revolution of chicken meat consumption."

(p17 cage broiler producer)

**3.4.5 Perceptions of chickens.**　Considering the low priority given to animal welfare in production, it is not surprising that most producers perceived chickens as tools of production or equalised animals as animal-based protein products. For example, producers explicitly stated how they perceived chickens:

"The role of chickens is a production tool. Essentially, it is for providing protein for humans."

(P13 cage and cage-free egg producer)

"Nowadays broilers have become an irreplaceable source of meat products in China."

(P22 cage and cage-free broiler producer)

That said, a couple of cage-free producers expressed some sympathy for chickens:

"I sympathise with farm animals. . .after a long time being with the hens, we feel an attachment to them. . . animals deserve some freedom during a short life cycle. . .I think it is good to provide some happiness and space while they produce."

(P15 cage-free egg producer)

"For broilers, although they are raised on the littered floor, I feel they are miserable because they can't see the sunlight even once."

(P30 cage-free broiler producer)

A small number of producers saw laying hens as sentient individuals as well as production tools:

"Laying hens are more intelligent than I thought. You can see their cognitive capabilities, temperament, personalities and likes and dislikes. . .a flock of laying hens are like a factory. . .just like machines. . .but they also have emotions. . .while achieving their production performance, we can minimise their suffering."

(P6 cage and cage-free egg producer)

"When I raised the first flock of birds, I saw them as my babies, and I was not dared to eat them and sold them. . .later on, I feel [raising broilers] is providing animal protein products for humans. . .now I feel [the broilers] are just healthy animal protein."

(P27 cage and cage-free broiler producer)

Participants' attitudes towards animal welfare were inextricably linked to economic considerations. Animal welfare was not a key consideration in production, and producers prioritised financial success in the market. Broiler producers were more likely to oppose applying Western animal welfare standards in a Chinese context and perceived animal welfare as a Western political strategy to hinder China's development. While there appeared to be both a caring and an instrumental relationship between animals and producers, producers stressed animals' instrumental values more.

## 4. Discussion

### 4.1 Producers' knowledge and definitions of the term animal welfare

The study investigated Chinese poultry producers' knowledge and understanding of animal welfare, explored the reasons for selecting housing systems, and revealed their attitudes toward animal welfare in poultry production. The findings demonstrate participants' familiarity with the term "animal welfare" and their strong grasp of its definitions. Some participating producers even mentioned "animal welfare" spontaneously and used it interchangeably with cage-free egg production systems, indicating their comfort with the phrase. Recent research suggests increased awareness and knowledge of animal welfare among Chinese producers [50, 52]. However, those outside the farming leadership in China, such as dairy farm workers [45, 46], workers in the transportation and slaughter sectors [63], and the public [64, 65], remain less acquainted with the concept.

Improved access to information and promotion of animal welfare in the livestock industry has raised awareness among producers. Animal welfare science was initially introduced to China's livestock industry in the early 1990s to address animal health and food safety concerns in intensive production systems [66]. Additionally, import restrictions on certain Chinese animal-based products in the European market compelled the Chinese government to emphasise animal welfare [67]. Various initiatives, such as research projects, conferences, training, and workshops, were implemented to heighten awareness and promote higher welfare practices in the livestock industry [66–68]. The participants' improved education levels and job statuses

likely facilitated access to these knowledge-sharing opportunities. Similar to prior research [46, 50, 52], business leaders in this study exhibited notable awareness of animal welfare.

The study's findings highlight participants' understanding of animal welfare concepts commonly defined in Western literature, in particular the three dimensions of animal welfare (i.e. biological functioning, affective states and natural behaviour) [32] or Five Freedoms [69]. This familiarity is attributed to the significant influence of Western organisations and institutions in disseminating these concepts in China through various means [68], including policy development [70], knowledge transfer [66, 68], research funding [71], and modelling higher welfare farming practices [72]. Despite not using the term, China has a rich history of poultry production spanning thousands of years [3], and the idea of animal welfare aligns with Chinese traditional culture [73]. Confucianism, Taoism, and Buddhism promote compassion towards animals and harmonious human and non-human animal relationships. The study underscores that familiarity with the term is not a prerequisite for implementing higher welfare practices. For instance, certain Chinese dairy farm workers unfamiliar with the term "animal welfare" were still knowledgeable about cattle welfare based on their daily tasks [46]. Similarly, participants in this study grasp the underlying conceptual and practical facets of animal welfare without a native definition.

In alignment with prior European literature [41, 74–76], participants demonstrated heterogeneity in their focused facets of animal welfare and attitudes toward different housing systems. Egg producers prioritised behavioural expression and supported cage-free housing, while broiler producers held opposing views. Animal welfare literature often highlights a disparity between minimising health issues and promoting natural behaviours [77]. Stakeholders attribute varying importance to these dimensions in their welfare assessments [78]. Earlier studies indicate that conventional producers prioritise health over natural behaviours [35, 38, 49, 50, 79], whilst users of higher welfare systems stress animals' natural behaviours [36, 44, 80, 81]. Given the diversity among participants, transitioning to and maintaining cage-free systems necessitates a tailored approach, as universal strategies may prove less effective than individualised tactics.

Participants' pronounced emphasis on animal health can be linked to various factors, including concerns about food safety and challenges in mitigating animal health issues within China. Animal farming in China is susceptible to food safety issues and potential public health hazards due to the widespread misuse and overuse of veterinary drugs, particularly antimicrobials, leading to the propagation of antimicrobial resistance (AMR) from animals to humans [82]. As one of the largest global consumers of antimicrobials [83], China employs antimicrobials extensively in animal agriculture [82], hence producers are anticipated to improve animal health management (such as improved hygiene in housing environments) to diminish antimicrobial use [82]. However, compared to other countries, China's disease control system has weaknesses, including unclear policies, ineffective enforcement, limited capacity among workers, resource shortages like vaccinations, and insufficient knowledge of on-farm biosecurity practices [49, 84]. Therefore, animal health is among the primary challenges when producers consider improving animal welfare on farms [50] and transitioning to cage-free egg production [22].

## 4.2 Producers' perceived priorities in considering housing systems

This study reveals that external factors beyond personal values significantly influence producers' housing system decisions. These choices hinge on perceived profitability, market demand, organisational leadership and policies. Previous research shows that Western producers' housing system choices are affected by external factors such as mandatory legislation [37, 85],

resource availability [75], and economic returns [75, 76, 86]. Western farmers are willing to adopt higher welfare production systems if they can gain price premiums to offset the increased costs [35–38, 87, 88]. Similarly, Chinese producers are driven mainly by the profitability and marketability of products to change to higher welfare farming practices [39, 52].

This study's findings indicate that companies' leadership, policies, and culture significantly influence the selection and maintenance of cage-free systems. Chinese cultural values, such as the paternalistic management style and acceptance of hierarchy, primarily influence Chinese companies [89]. Earlier research reveals the importance of Chinese companies' leadership, policies, and organisational culture in motivating and supporting workers to improve animal welfare in China [45, 46, 53]. Participating leaders in this study prioritised profitability and market success. Thus, transparent communication regarding the commercial advantages of improving animal welfare might be a pivotal strategy for engaging with these business leaders.

Certain participants in this study who endorse the benefits of cage-free systems for animal welfare showed hesitance in transitioning to such systems. This hesitancy corresponds to the intricate relationship between productivity, profitability, and animal welfare. While some studies indicate better egg and meat performance in free-range systems [90, 91], others find cage-free systems comparable or inferior in meat and egg production [92, 93]. China-based research suggests variable productivity and profitability outcomes, with some favouring cage systems [5, 94, 95] and others supporting free-range production [24, 96]. Similar debates extend to other regions where cage-free systems have been employed [22, 97–99], yet data primarily originate from experimental or small-scale farms, necessitating economic evidence from large cage-free operations. While Chinese consumers are willing to pay for higher welfare products [47, 100, 101], translating intentions into actions lacks empirical confirmation. Chinese livestock stakeholders, therefore, demand evidence-based economic results [30, 52] due to scepticism about reconciling increased profits with elevated animal welfare costs [39]. Since cage-free adoption raises costs, entails substantial capital investment [102–104], and introduces market risks from uncertain demand and fluctuating prices [105], economic considerations play a more critical role in deciding on housing systems than moral judgment.

Therefore, this study verifies that economic motives precede animal welfare arguments when producers choose higher welfare housing systems. While knowledge and attitudes might align with behaviours, they are not the most critical determinants of actions. Instead, external factors largely shape farmers' behaviours [106]. Highlighting this gap between knowledge and action is significant, suggesting that knowledge alone may not hinder improving animal welfare in China, with market incentives potentially proving more effective in engaging producers.

## 4.3 Producers' attitudes toward animal welfare

Several participants in this study did not prioritise animal welfare, expressing a preference for cages in poultry production. This attitude aligns with previous research that alternative factors (e.g., food safety, food security, and profits) outweigh animal welfare for Chinese farmers [39, 45, 52]. The industry's perceived low emphasis on animal welfare can be traced to China's political, socio-economic, and cultural contexts, as observed in other regions [36, 39, 107].

At the political level, the Chinese government has prioritised scaling up and intensifying animal farming to expedite agricultural development to alleviate poverty, address food shortages [108], and enhance food safety. In China, food supply underpins social stability, as captured by the saying: "王者以民为天, 民以食为天" (people are the most important to an emperor, and food is the most important to the people). Moreover, export rejections in international markets and domestic food safety scandals have heightened awareness and concerns

about food safety among Chinese consumers [109, 110]. Consequently, China welcomes industrialised and intensive production systems, such as battery cages [111], seen as "scientific" tools to boost poultry output [43, 111] and enhance food risk management [96].

Furthermore, China's lack of legislation and limited public engagement contributes to the lower priority of animal welfare in poultry production. Driven by animal welfare science, advocacy groups [5, 14–16], and corporates' commitments to source higher welfare eggs and chicken meat [20], conventional cages have been banned for laying hens in Europe [97, 112]. In contrast, China lacks national animal welfare laws despite attempts to establish animal protection regulations [73]. While the Chinese government shows willingness for animal welfare improvement, it allows limited public engagement in animal welfare initiatives due to the term's associations with animal and human rights [67] and concerns that public activism might jeopardise political stability [71]. Moreover, animal agricultural industries are antagonistic toward international advocacy groups and their pressure to improve animal welfare because the industry might feel their cultural identities are under attack [29]. As such, similar to previous research [30], participants in this study demanded that farm animal welfare standards be developed and applied in the Chinese context rather than copying Western standards, or as Participant 17 put it: "We must [develop] animal welfare with Chinese characteristics".

At the social level, Chinese consumers' limited awareness of animal welfare and reluctance to buy cage-free products discourage producers from prioritising animal welfare in production. In Europe, concerns about animal welfare, the environment, and food safety lead Western producers to hesitate to scale production [79]. In China, animal welfare awareness for farmed animals is lower than for companion animals and wildlife [64, 65], and Chinese consumers prioritise food safety and support industrial farms [113]. Nevertheless, recent studies show that educated, higher-income young consumers are willing to pay more for higher animal welfare products [47, 114]. The approaching deadline for the commitment to cage-free eggs by 2025 puts pressure on food businesses to source cage-free eggs [115]. This suggests that Chinese buyers could drive producers to adopt high-welfare practices and cater to a future niche market [67].

The perceived lower importance of animal welfare in production among the producers is related to the perception of chickens as commodities lacking sentience. In Chinese culture, animals are predominantly viewed as resources [67]. This viewpoint posits that the concept of animal welfare, rooted in Western values, could be deemed too forward-thinking for Chinese society [73]. "Welfare" is often understood in the context of human welfare [73] and seen as "luxurious propaganda," leading to resistance against animal welfare legislation [68, 116] and reluctance to promote it until human rights and welfare are fully realised [68].

The limited emphasis on animal welfare and sentience might also stem from farmer-animal relationships. Participants' leadership roles often disconnect them from daily animal care, and large flock sizes hinder individual connections with chickens. Moreover, compared to other species (e.g. cows), producers might feel less attachment to chickens due to their brief farm stays (particularly white-feathered broilers) and the purpose of raising them for slaughter. This physical and emotional distance can foster an "attitude of detached detachment" [117] (p. 219). Consequently, some producers with daily animal contact display empathy, while many perceive animals as protein products, which is often seen in commercial settings [117]. Detached human-animal relationships on Chinese farms suggest that advocating animal welfare on ethical grounds might not effectively engage poultry producers. Instead, identifying links between animal welfare, food safety, marketability, and profitability could be a more strategic approach to promoting cage-free production in China.

Nevertheless, some egg producers demonstrated strong intentions to adopt cage-free housing systems. Contextual differences in the egg and broiler sectors might explain the higher

intention to go cage-free among the egg producers than the broiler producers. In Europe, the cage-free egg-sourcing commitment started in 2007 [118]. In contrast, the Better Chicken Commitment requiring non-cage systems in the broiler industry only began in 2016, and the companies who made Better Chicken commitments are limited to their western markets such as Europe, America and Australia [119]. As a result, similar to Europe, while the demand for cage-free eggs and discussion of transitioning to cage-free systems has been escalating in China [23, 27], less attention is paid to broiler welfare in cages than the concerns over cages for laying hens in academia [16] and food businesses in China [119]. Therefore, Chinese egg producers were more driven to switch to cage-free systems than broiler sector producers.

## 5. Conclusion

Based on interviews with thirty Chinese poultry producers employing cage and cage-free systems, this study highlights producers' generally robust awareness and knowledge of animal welfare. While health and productivity, affective state, and natural behaviours collectively shape their understanding of animal welfare, egg producers prioritise natural behaviours, while broiler producers emphasise health and productivity. In their housing system choices, external factors such as profitability, company leadership, and organisational policies wield greater influence than animal welfare considerations. These producers do not accord primary importance to animal welfare in poultry production and view chickens more as production tools than sentient beings.

This study offers insights into designing strategies to encourage Chinese poultry producers to adopt cage-free systems and enhance animal welfare. Despite heightened awareness of animal welfare, producers' perceptions and knowledge do not determine housing system choices. Instead, their attitudes and behaviours towards animal welfare are intertwined with company, sector, and China-specific contexts. Moreover, egg producers exhibit a stronger inclination to transition to cage-free systems, driven by potential market demand for cage-free eggs compared to the broiler sector. Consequently, prioritising engagement with egg producers for cage-free adoption is a strategic approach, with learnings applicable to the broiler sector in the future. Given China's economic, political, social, and cultural milieu, this study highlights market incentives as a potentially effective strategy for promoting cage-free systems, necessitating engagement with corporate buyers to expedite the shift. However, comprehensive strategy development requires a more extensive exploration of Chinese egg producers' and corporations' perceived opportunities and challenges in the transition process.

## Supporting information

**S1 Appendix. Interview guide for egg producers.**
(DOCX)

**S2 Appendix. Interview guide for broiler producers.**
(DOCX)

## Acknowledgments

The authors thank all the egg and broiler producers interviewed for their time and participation.

## Author Contributions

**Conceptualization:** Qing Yang, Belinda Vigors.

**Data curation:** Qing Yang.

**Formal analysis:** Qing Yang.

**Funding acquisition:** Cathy M. Dwyer, Fritha M. Langford.

**Methodology:** Qing Yang.

**Supervision:** Cathy M. Dwyer, Belinda Vigors, Ruqian Zhao, Fritha M. Langford.

**Writing – original draft:** Qing Yang.

**Writing – review & editing:** Cathy M. Dwyer, Belinda Vigors, Fritha M. Langford.

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
