## [Decision Letter · Decision Letter 0]

11 Mar 2024

PONE-D-24-01570Animal welfare with Chinese characteristics: Chinese poultry producers’ perceptions of, and attitudes towards, animal welfarePLOS ONE

Dear Dr. Yang,

Thank you for submitting your manuscript to PLOS ONE. After careful consideration, we feel that it has merit but does not fully meet PLOS ONE’s publication criteria as it currently stands. Therefore, we invite you to submit a revised version of the manuscript that addresses the points raised during the review process.

We look forward to receiving your revised manuscript.

Kind regards,

Lamiaa Mostafa Radwan, Ph.D.

Academic Editor

PLOS ONE

Journal Requirements:

"The authors thank all the egg and broiler producers interviewed for their time and participation. This work was supported by the Open Philanthropy Project"

"Dr C.M.D. and the Royal (Dick) School of Veterinary Studies at University of Edinburgh received funding for this project through Open Philanthropy (https://www.openphilanthropy.org/). There is no grant number for this award. The funder had no role in the study design, data collection and analysis, publication decision, and manuscript preparation."

**Additional Editor Comments:**

Dear Dr., Qing Yang

Thank you for submitting your manuscript to PLOS ONE. After careful consideration, we have decided that your manuscript needs Major Revision.

Kind regards,

Prof. Lamiaa Mostafa Radwan, Ph.D.

Academic Editor

PLOS ONE

Reviewer1

This is a carefully done study and the findings are of considerable interest. However, some points need clarifying and certain statements require further justification. There are given below.

1. What is the basis of the interview guide? Do you refer to previous studies? Line 175 seems to mention some references, but it still raises my concerns about the specifics, such as how do you modify the question to fit your research? How can the question be modified so that the questions asked of employees in the reference apply to managers? The lines 154 and 175, which appear to be related to the questionnaire, are not put together.

2. More details about recruiting participants are needed. Why did these farm owners, managers, or senior leaders volunteer to participate in your research? The current statement is untenable.

3. Is the farm size the full size of the participants’ company or the full size they manage? The current one is ambiguous (table 1 and 2). Please add more farm details like Chen [45] did, including available basic demographics and operating characteristics.

4. line 152: the of was set to be upper corner format

5. The discussion is large and interesting. However, appropriate subheadings may give the reader a clearer picture of the structure of the discussion. It is necessary to remove some irrelevant redundant content.

6. There is one thing that puzzles me. Are any of the 30 respondents from the same company or the same person? Are there cases where companies raise both layers and broilers? If not, have you deliberately avoided such participants?

7. Since the participants are at a high level in the organization, their answers may be more official or crowning. Even their answers are not what they really think, but what the company tells them to say. Analyzing such answers is pointless and can lead to results that are significantly biased towards reality. How do you consider and address this issue?

8. Finally, I must emphasize that your title is about PRODUCER and the participants in the manuscript are FARM OWNERS, MANAGERS, or SENIOR LEADERS, who are not producers in the traditional sense of the word (i.e., direct participants in the practice of poultry production). Therefore, the word PRODUCER in the title could be replaced with another word that better summarizes the participants in this study.

If all my technical concerns are addressed, this manuscript could be considered for publication.

Reviewer2

I enjoyed reading this manuscript. The study appears to be well-conducted and the paper is very well written, describing the findings in an impactful way. A lot of interesting information arose from these interviews and the study will make an important contribution to the field, especially for those working cross-culturally.

I have a few suggestions to consider:

L50: "Intensive nature". What is described here is really the scale rather than intensity

L52-54. This statement suggests that most production of eggs and chicken meat in China is for the domestic market. Is that the case?

L58. "While various poultry" suggest to change to "while all poultry..."

L67. What makes these companies "leading". Who are they? Is it marketshare or another factor?

L82. A reference is needed for the "billions" statement. So far the level of production has been measured in terms of production units e.g. eggs / tonnes of meat

L92. This statement "individuals' diverse perspectives on ethical animal treatment" is a bit vague and can be improved

L94. For this statement "different degrees of significance attached to animal welfare in animal production" - do you mean in terms of competing priorities?

L96 - 97. For the aspects mentioned here e.g. "country-specific dynamics", "sector-specific considerations", "farm-specific contexts" these statements are a bit vague. It would be good to add a few examples after each in brackets so that what you are referring to is more evident.

L111. Suggest change "welfare in their practices" might be better as "welfare in practice"

L134-137. It would be good to define/describe these systems for readers not in this industry or from a region that uses different terminology

L146. It would be good to know more about these participants. I assume some information was collected on them outside of what is described here. For example, how long have they worked in the industry, level of education etc. In essence, more information is needed so the reader can understand how much their opinions and statements hold weight.

Tables 1 & 2, ensure the Chinese names are consistently romanised.

166. Be more specific for this statement "checked some quotes".

L178. Were the participants ever given a standardised definition of animal welfare?

L514. Typo - "anima" should be "animal"

Reviewers' comments:

Reviewer's Responses to Questions

**Comments to the Author**

1. Is the manuscript technically sound, and do the data support the conclusions?

Reviewer #1: Partly

Reviewer #2: Yes

2. Has the statistical analysis been performed appropriately and rigorously? 

Reviewer #1: N/A

Reviewer #2: N/A

3. Have the authors made all data underlying the findings in their manuscript fully available?

Reviewer #1: Yes

Reviewer #2: No

4. Is the manuscript presented in an intelligible fashion and written in standard English?

Reviewer #1: Yes

Reviewer #2: Yes

5. Review Comments to the Author

Reviewer #1: This is a carefully done study and the findings are of considerable interest. However, some points need clarifying and certain statements require further justification. There are given below.

1. What is the basis of the interview guide? Do you refer to previous studies? Line 175 seems to mention some references, but it still raises my concerns about the specifics, such as how do you modify the question to fit your research? How can the question be modified so that the questions asked of employees in the reference apply to managers? The lines 154 and 175, which appear to be related to the questionnaire, are not put together.

2. More details about recruiting participants are needed. Why did these farm owners, managers, or senior leaders volunteer to participate in your research? The current statement is untenable.

3. Is the farm size the full size of the participants’ company or the full size they manage? The current one is ambiguous (table 1 and 2). Please add more farm details like Chen [45] did, including available basic demographics and operating characteristics.

4. line 152: the of was set to be upper corner format

5. The discussion is large and interesting. However, appropriate subheadings may give the reader a clearer picture of the structure of the discussion. It is necessary to remove some irrelevant redundant content.

6. There is one thing that puzzles me. Are any of the 30 respondents from the same company or the same person? Are there cases where companies raise both layers and broilers? If not, have you deliberately avoided such participants?

7. Since the participants are at a high level in the organization, their answers may be more official or crowning. Even their answers are not what they really think, but what the company tells them to say. Analyzing such answers is pointless and can lead to results that are significantly biased towards reality. How do you consider and address this issue?

8. Finally, I must emphasize that your title is about PRODUCER and the participants in the manuscript are FARM OWNERS, MANAGERS, or SENIOR LEADERS, who are not producers in the traditional sense of the word (i.e., direct participants in the practice of poultry production). Therefore, the word PRODUCER in the title could be replaced with another word that better summarizes the participants in this study.

If all my technical concerns are addressed, this manuscript could be considered for publication.

Reviewer #2: I enjoyed reading this manuscript. The study appears to be well-conducted and the paper is very well written, describing the findings in an impactful way. A lot of interesting information arose from these interviews and the study will make an important contribution to the field, especially for those working cross-culturally.

I have a few suggestions to consider:

L50: "Intensive nature". What is described here is really the scale rather than intensity

L52-54. This statement suggests that most production of eggs and chicken meat in China is for the domestic market. Is that the case?

L58. "While various poultry" suggest to change to "while all poultry..."

L67. What makes these companies "leading". Who are they? Is it marketshare or another factor?

L82. A reference is needed for the "billions" statement. So far the level of production has been measured in terms of production units e.g. eggs / tonnes of meat

L92. This statement "individuals' diverse perspectives on ethical animal treatment" is a bit vague and can be improved

L94. For this statement "different degrees of significance attached to animal welfare in animal production" - do you mean in terms of competing priorities?

L96 - 97. For the aspects mentioned here e.g. "country-specific dynamics", "sector-specific considerations", "farm-specific contexts" these statements are a bit vague. It would be good to add a few examples after each in brackets so that what you are referring to is more evident.

L111. Suggest change "welfare in their practices" might be better as "welfare in practice"

L134-137. It would be good to define/describe these systems for readers not in this industry or from a region that uses different terminology

L146. It would be good to know more about these participants. I assume some information was collected on them outside of what is described here. For example, how long have they worked in the industry, level of education etc. In essence, more information is needed so the reader can understand how much their opinions and statements hold weight.

Tables 1 & 2, ensure the Chinese names are consistently romanised.

166. Be more specific for this statement "checked some quotes".

L178. Were the participants ever given a standardised definition of animal welfare?

L514. Typo - "anima" should be "animal"

6. PLOS authors have the option to publish the peer review history of their article (what does this mean?). If published, this will include your full peer review and any attached files.

Reviewer #1: No

Reviewer #2: **Yes: **Kris Descovich

---

## [Author Response · Author response to Decision Letter 0]

27 May 2024

Reviewer1

1. What is the basis of the interview guide? Do you refer to previous studies? Line 175 seems to mention some references, but it still raises my concerns about the specifics, such as how do you modify the question to fit your research? How can the question be modified so that the questions asked of employees in the reference apply to managers? The lines 154 and 175, which appear to be related to the questionnaire, are not put together.

We did not refer to previous studies to develop the interview guides used in this study. It is the first of its kind in a Chinese context, and we have embarked on an exploratory approach to capture rich and nuanced insights. The interview guides were developed iteratively, drawing upon the expertise of our research team and consulting with industry professionals to ensure the relevance and appropriateness of the questions. The interview questions were guided by the overarching research questions, which aimed to investigate producers' knowledge and perceptions of animal welfare, decision-making processes regarding housing systems and the importance of animal welfare in poultry production. In the meantime, the authors also conducted a literature review related to producers’ perspectives on animal welfare and/or poultry housing systems. While this literature review informed our understanding of key concepts and potential areas of inquiry, it did not serve as a direct template for our interview questions. By adopting an iterative approach to developing our interview guide, we explored the topics of interest in-depth while allowing flexibility for participants to share their perspectives and experiences. 

In response to the query regarding modifying interview questions to suit participants in employee and leadership positions, we would like to clarify that all the participants occupy leadership roles. Our recruitment criteria specifically targeted individuals in leadership positions to understand the decision-making processes. Moreover, our interviews were conducted using a semi-structured format, allowing for adaptability and flexibility in questioning. Instead of rigidly predefined questions, this approach enables the interviewer to explore relevant topics based on the participant's answers, tailored to each participant's unique circumstances and responses. Lines 138-140 specified the targeted population.

2. More details about recruiting participants are needed. Why did these farm owners, managers, or senior leaders volunteer to participate in your research? The current statement is untenable.

All participants in our study volunteered to participate, suggesting an interest in contributing to the research. Before their involvement, participants were provided with an information sheet detailing the purpose of the study, the procedures involved, and their rights as participants. They were also asked to sign a consent form affirming their voluntary participation and understanding of the study's objectives and procedures. The participants were assured they could withdraw from the study without facing any negative consequences.

We did not explicitly inquire about the participants’ intention to participate because it was not directly relevant to our research questions. Instead, we focused on eliciting insights into participants' perceptions and attitudes towards animal welfare in poultry production, their decision-making processes related to housing systems and the importance of animal welfare. Moreover, we chose not to provide incentives for participation to avoid any potential influence on participants' motivations or responses. We contacted 34 potential participants, of whom four declined to participate in the interviews. While we did not inquire about the specific reasons for their rejection, we respected their decision without imposing any pressure or follow-up.

Therefore, the recruitment process was conducted following ethical guidelines and standards, and we focused on voluntary participation, informed consent, and protecting participants' identities. 

3. Is the farm size the full size of the participants’ company or the full size they manage? The current one is ambiguous (table 1 and 2). Please add more farm details like Chen [45] did, including available basic demographics and operating characteristics.

The farm size described in Tables 1 and 2 reflects the scale of operations the participants manage or supervise. For some participants, the entire company’s farms are under the participants’ management or supervision. 

In adherence to ethical guidelines and to ensure participant confidentiality and anonymity, our data collection process was designed to collect minimal personal and farm-related information, as approved by the Human Ethical Review Committee. Specifically, we only recorded participants' job positions, the size of the farm(s) they manage, and the provincial location of the farm. These details were essential for confirming participants' eligibility based on our recruitment criteria, outlined in lines 138-145.

The primary focus of our study is to explore participants' perspectives and attitudes toward animal welfare and poultry housing systems. As such, our research objectives do not extend to investigating how demographic or operational factors, such as age, gender, education, work experience, or farm history, influence these perspectives. Therefore, we did not collect additional demographic or operational data beyond what was necessary for participant selection and contextual understanding.

In summary, the limited scope of our data collection aligns with ethical considerations and the specific focus of our research objectives. While we appreciate the suggestion to include additional farm details, as demonstrated in Chen [45], the nature of our study and ethical constraints preclude us from collecting and providing further information on demographics and operating characteristics.

4. line 152: the of was set to be upper corner format

The mistake has been corrected. Please see line 172.

5. The discussion is large and interesting. However, appropriate subheadings may give the reader a clearer picture of the structure of the discussion. It is necessary to remove some irrelevant redundant content.

Three subheadings have been added under the heading of section “5 Discussion”:

5.1 Producer’s knowledge and definitions of the term animal welfare

5.2 Producers’ perceived priorities in considering housing systems

5.3. Producer’s attitudes toward animal welfare

Some sentences in the discussion section have been edited or removed. For instance, line 674-681, the original text, “compared to many other countries, some specific weakness in China’s disease control system reportedly results in challenges in ensuring animal health, including a lack of clear policies and effective enforcement of regulations, low capacity of disease control workers, shortage of resources (e.g., vaccinations), and insufficient knowledge of on-farm biosecurity practices” has been edited to a more concise sentence “compared to other countries, China's disease control system has weaknesses, including unclear policies, ineffective enforcement, limited capacity among workers, resource shortages like vaccinations, and insufficient knowledge of on-farm biosecurity practices”. Some redundant information has been deleted, for example, line 752-753 “animal welfare groups in China often adopt a collaborative instead of a confrontational approach as the government supervises their operations”. Please see the file “revised manuscript with track changes”. 

6. There is one thing that puzzles me. Are any of the 30 respondents from the same company or the same person? Are there cases where companies raise both layers and broilers? If not, have you deliberately avoided such participants?

All respondents represent 30 companies within the poultry sector, none of which raise both layers and broilers. This separation arises due to differences in species, production processes, and biosecurity regulations. While both sectors (i.e. laying hen and broiler) have cages and cage-free housing systems, they require distinct feed, technologies, and husbandry practices. Additionally, companies targeting egg production and chicken meat operate in separate markets and distribution networks, contributing to the sectors' distinct trajectories in modern agriculture. Our study aims to explore producers' perceptions of animal welfare in relation to cages and cage-free systems across these sectors, hence the inclusion of both types of farms. Given the lack of companies raising both layers and broilers, this participant group was not deliberately avoided. We focus on capturing insights from producers within each sector to understand their perspectives on and attitudes toward animal welfare and housing systems.

7. Since the participants are at a high level in the organization, their answers may be more official or crowning. Even their answers are not what they really think, but what the company tells them to say. Analyzing such answers is pointless and can lead to results that are significantly biased towards reality. How do you consider and address this issue?

We appreciate the reviewer's concerns over the social desirability bias. In our research, we recognise the potential for social desirability bias, where participants may feel inclined to provide responses they perceive as socially favourable or expected rather than expressing their genuine thoughts or feelings, particularly given the high-level positions of our participants within their organisations. This bias may occur when collecting data on sensitive topics, such as animal welfare in China's animal agriculture, where societal expectations may influence responses. To mitigate this bias, we selected participants involved in decision-making processes related to both cage and cage-free systems to capture diverse perspectives and motivations. Before and during the interviews, we assured participants of confidentiality and anonymity to encourage candid responses during interviews. Moreover, we carefully designed interview questions to avoid leading prompts that could shape participants' responses according to perceived research expectations.

Despite assumptions that favouring animal welfare and endorsing cage-free systems are socially desirable responses, our analysis revealed that animal welfare considerations were not consistently prioritised in the decision-making processes of poultry production. Hence, the social desirability bias is not as prevalent as expected. 

8. Finally, I must emphasize that your title is about PRODUCER and the participants in the manuscript are FARM OWNERS, MANAGERS, or SENIOR LEADERS, who are not producers in the traditional sense of the word (i.e., direct participants in the practice of poultry production). Therefore, the word PRODUCER in the title could be replaced with another word that better summarizes the participants in this study.

We appreciate the reviewer's attention to detail regarding the terminology used in our title. Whilst our participants may not directly engage in the physical aspects of poultry production, they play crucial roles in decision-making processes, resource allocation, and policy formulation at the production stage. Thus, the term 'producer' encompasses the broader spectrum of individuals managing and overseeing poultry production operations, including farm owners, managers, and senior executives. However, we acknowledge that some readers might find the term 'producer' misleading. To address this concern, we added clarifying information in the Abstract to emphasise that our participants are not solely individuals directly involved in poultry production practices but also those in managerial and supervising roles (lines 26-28). Additionally, we have included similar information in the method section to specify that “the participants comprised professionals in different roles within farming companies, including farm owners, managers, and senior executives who oversee one or multiple farms in large integrated animal production firms. For the sake of brevity and simplicity in the following sections, these individuals are collectively referred to as producers” (lines 161- 165). We believe that retaining the term 'producer' in the title and additional specifications in the abstract and method sections can reflect the essence of our study and the participants' roles.

Reviewer 2

1. L50: "Intensive nature". What is described here is really the scale rather than intensity

We appreciate the reviewer's insightful observation. Upon reviewing the text, we agree that the term 'intensive' may not be accurate in the context characterised more by its large-scale operations rather than intensity. We have revised the text accordingly to state that “China's poultry sector is characterised by its growing scale of operations”. Please see line 51.

2. L52-54. This statement suggests that most production of eggs and chicken meat in China is for the domestic market. Is that the case?

According to the references cited, the implication that most eggs and chicken meat produced in China is for domestic consumption is valid. 

The statement that most production of eggs and chicken meat in China is for the domestic market is based on the references cited in this sentence, “This trajectory of intensification and industrialisation will persist to meet the escalating demand for eggs and poultry meat from the country's vast population [3, 8]”. Citation 3 says, “Egg exports are also low, about 0.25% of the total production” (p32), indicating that most eggs produced in China are consumed domestically rather than exported. In citation 8, it says, “As the second most highly consumed meat in China (second only to pork), …for the future, it is expected that the meat product demand will continue to rise as a result of a growing population, increased urbanization, and income growth both in the urban and rural areas” (p2). Citation 8 emphasises the high consumption of chicken meat domestically, highlighting it as the second most consumed meat in China after pork. This suggests that the demand for chicken meat is primarily driven by domestic consumption trends, with broiler meat significantly contributing to providing animal protein for urban and rural residents. Given the expected rise in meat product demand due to population growth, urbanisation, and income growth, it can be argued that the primary focus of egg and chicken meat production in China is to cater to domestic market needs.

3. L58. "While various poultry" suggest to change to "while all poultry..."

“various” has been changed to “all”. Please see line 59.

4. L67. What makes these companies "leading". Who are they? Is it marketshare or another factor?

The authors agree that the word “leading” is unclear in the sentence: “Globally, numerous leading companies in the food industry have committed to sourcing cage-free eggs from suppliers by 2025, initially focusing on European and North American markets.” To improve its clarity according to the cited reference, the sentence has been changed to “Dozens of global manufacturers, hotels, supermarket chains, and food service companies have committed to sourcing cage-free eggs throughout their international operations, initially focusing on European and North American markets.” Please see lines 70-72.

5. L82. A reference is needed for the "billions" statement. So far the level of production has been measured in terms of production units e.g. eggs / tonnes of meat.

The authors agree that a reference is needed for the “billions” statement. According to FAO Stats, the number of chickens used for eggs and meat was over 5.2 billion. Hence, the reference is added to the text. Please see line 87, where the reference is inserted.

6. L92. This statement "individuals' diverse perspectives on ethical animal treatment" is a bit vague and can be improved

The authors agree that the wording of "individuals' diverse perspectives on ethical animal treatment" is vague, and the text has been improved: “In addition, Western producers' decisions regarding production systems may be influenced by diverse perspectives on ethical animal treatme

---

## [Decision Letter · Decision Letter 1]

1 Jul 2024

Animal welfare with Chinese characteristics: Chinese poultry producers’ perceptions of, and attitudes towards, animal welfare

PONE-D-24-01570R1

Dear Dr. Yang,

We’re pleased to inform you that your manuscript has been judged scientifically suitable for publication and will be formally accepted for publication once it meets all outstanding technical requirements.

Kind regards,

Lamiaa Mostafa Radwan, Ph.D.

Academic Editor

PLOS ONE

Additional Editor Comments (optional):

Accept

Reviewers' comments:

Reviewer's Responses to Questions

**Comments to the Author**

1. If the authors have adequately addressed your comments raised in a previous round of review and you feel that this manuscript is now acceptable for publication, you may indicate that here to bypass the “Comments to the Author” section, enter your conflict of interest statement in the “Confidential to Editor” section, and submit your "Accept" recommendation.

Reviewer #1: All comments have been addressed

2. Is the manuscript technically sound, and do the data support the conclusions?

Reviewer #1: Yes

3. Has the statistical analysis been performed appropriately and rigorously? 

Reviewer #1: N/A

4. Have the authors made all data underlying the findings in their manuscript fully available?

Reviewer #1: (No Response)

5. Is the manuscript presented in an intelligible fashion and written in standard English?

Reviewer #1: Yes

6. Review Comments to the Author

Reviewer #1: (No Response)

7. PLOS authors have the option to publish the peer review history of their article (what does this mean?). If published, this will include your full peer review and any attached files.

Reviewer #1: **Yes: **Bing Jiang

---

## [Editor Report · Acceptance letter]

8 Jul 2024

PONE-D-24-01570R1 

PLOS ONE

Dear Dr. Yang, 

I'm pleased to inform you that your manuscript has been deemed suitable for publication in PLOS ONE. Congratulations! Your manuscript is now being handed over to our production team.

Kind regards, 

on behalf of

Prof. Dr. Lamiaa Mostafa Radwan 

Academic Editor

PLOS ONE